FEM: mining biological meaning from cell level in single-cell RNA sequencing data

Liu Yunqing 1
Lu Na 1
Bi Changwei 1 2
Han Tingyu 1
Zhuojun Guo 1
Zhu Yunchi 1
Li Yixin 1
He Chunpeng cphe@seu.edu.cn 1
Lu Zuhong zhlu@seu.edu.cn 1
State Key Laboratory of Bioelectronics, School of Biological Science and Medical Engineering, Southeast University , Nanjin , Jiangsu , China
Nanjing Forestry University, School of Information Science and Technology , Nanjin , Jiangsu , China
Soares Paula
Electronic publication date: 2021 Nov 30
Publication date: 2021
Volume: 9
Electronic Location ID: e12570
Received 2021 Apr 27; Accepted 2021 Nov 8
Copyright: ©2021 Liu et al.
Copyright year: 2021
Copyright holder: Liu et al.
License: This is an open access article distributed under the terms of the Creative Commons Attribution License, which permits unrestricted use, distribution, reproduction and adaptation in any medium and for any purpose provided that it is properly attributed. For attribution, the original author(s), title, publication source (PeerJ) and either DOI or URL of the article must be cited.
License URL: https://creativecommons.org/licenses/by/4.0/

Keywords: Single-cell RNA sequencing, Gene set enrichment analysis, Functional expression matrix

Funding: National Science and Technology Major Project of China 6307030004 This work was supported by the National Science and Technology Major Project of China (6307030004). The funders had no role in study design, data collection and analysis, decision to publish, or preparation of the manuscript.

==============================
Background

One goal of expression data analysis is to discover the biological significance or function of genes that are differentially expressed. Gene Set Enrichment (GSE) analysis is one of the main tools for function mining that has been widely used. However, every gene expressed in a cell is valuable information for GSE for single-cell RNA sequencing (scRNA-SEQ) data and not should be discarded.

Methods

We developed the functional expression matrix (FEM) algorithm to utilize the information from all expressed genes. The algorithm converts the gene expression matrix (GEM) into a FEM. The FEM algorithm can provide insight on the biological significance of a single cell. It can also integrate with GEM for downstream analysis.

Results

We found that FEM performed well with cell clustering and cell-type specific function annotation in three datasets (peripheral blood mononuclear cells, human liver, and human pancreas).

Introduction

RNA sequencing (RNA-seq) has been widely used as an alternative to the microarray platform over the past 10 years. It has provided valuable insight into complex biological mechanisms, ranging from cancer genomics to diverse microbial communities. Single-cell RNA-seq (scRNA-seq) can facilitate new and potentially unexpected biological discoveries when compared with traditional bulk methods that profile batches of pooled cells. For example, it can provide information on complex and rare cell populations and regulatory relationships between genes and can track the trajectories of distinct cell lineages during development (Gawad, Koh & Quake, 2016; Hwang, Lee & Bang, 2018; Chen, Ning & Shi, 2019).

Cells are the fundamental unit in biology. Biologists have known for centuries that multicellular organisms are characterized by a plethora of distinct cell types (Kiselev, Andrews & Hemberg, 2019).

It is difficult to distinguish cell types from the genome sequence since all cells of an individual share the same set of genetic material. Their differences may be characterized by transcriptome similarity as defined by unsupervised clustering (Kiselev, Andrews & Hemberg, 2019).

Bulk transcriptome studies using samples containing multiple cells cannot distinguish between the cells within the sample. Single-cell transcriptomics can address this issue (Kulkarni et al., 2019). High-throughput single-cell transcriptomics have provided unprecedented insights into the cellular diversity of tissues across diverse organisms. scRNA-seq is a promising approach in the study of the transcriptomes of organism’s individual cells.

The data processing pipeline for scRNA-seq is based on bulk RNA-seq. The basic workflow includes quality control (Luecken & Theis, 2019; Ilicic et al., 2016; Griffiths, Scialdone & Marioni, 2018), normalization, data correction, feature selection, and dimensionality reduction, followed by cell-level and gene-level downstream analysis (Luecken & Theis, 2019). Many genes are expressed through RNA-seq data, which is made up of pooled samples of multiple cells. However, it remains unclear whether these genes are co-expressed or just the result of mixing. Therefore, a portion of the particularly well-characterized genes must be screened for downstream analysis.

This type of scRNA-seq data analysis has specific shortcomings. First, the characteristics of single-cell data are not fully utilized. scRNA-seq data are a direct reflection of the physiological state of a cell, but the aforementioned method is an analytical process based on multiple cells or cell clusters. Second, it cannot effectively capture all meaningful functional groups since many genes are filtered but may still play important roles in transient cell states. It is possible to miss meaningful biological functions since functional enrichment analysis is based on all of the cell’s expressed genes, including genes that were previously filtered out. Third, the results of downstream functional analysis and upstream clustering are not integrated and visualized. To overcome these limitations, this study proposes a method based on the functional expression matrix (FEM) which converts each cell’s gene expression into an enriched gene set representing a specific function.

Materials & Methods

Motivation for this approach

Functional Gene Set Enrichment Analysis (GSEA) is often the last step of expression data analysis. A gene set usually represents a biological function, and the function will be used to refer to a gene set in subsequent analyses. Many R packages and some online sites, such as DAVID, provide enrichment analysis tools. The software takes a set of genes given by the user as input and returns functions (pathways) that are significantly enriched. Enrichment analysis methods for individual samples have also been proposed (Foroutan et al., 2018; Hänzelmann, Castelo & Guinney, 2013).

Zhang et al. (2020) summarized the widely used GSEA algorithms. We divided their methods into three categories: methods based on dimension reduction, methods based on single samples, and methods based on phenotype or cluster.

Methods based on dimension reduction include Pagoda2 (Fan et al., 2016) and PLAGE (Tomfohr, Lu & Kepler, 2005). These methods convert the gene expression matrix into FEM by performing dimension reduction (SVD for PLAGE and PCA for Pagoda2) on each gene set (pathway). Dimension reduction uses all samples and is not entirely based on a single cell.

Methods based on phenotype or cluster include Vision (DeTomaso et al., 2019) and z-score (Lee et al., 2008). Vision methods clusters GEM into clusters by KNN, and GSEA is based on these clusters. The Z-score method depends on phenotypes, which define each pathway activity score based on the top gene of t-test score between two conditions. These gradually join the genes in pathways to maximize the activity score and are not based on a single sample.

Methods based on the single sample include AUCell (Aibar et al., 2017), ssGSEA (Barbie et al., 2009), and GSVA (Hänzelmann, Castelo & Guinney, 2013), which are all non-parametric test statistical methods based on gene rank. Data sparsity is the main problem when analysing data from a single cell. Therefore, the level of gene expression is not the primary consideration for a pathway, but rather, whether the gene is expressed. A method based on Fisher’s exact test or the chi-square test may have better results than the no-parameter test method based on ranking.

scRNA-seq has low capture efficiency and high dropouts due to the small amount of initial material used. scRNA-seq expression data tend to be sparse (Hicks et al., 2018; Eraslan et al., 2019). A true zero represents the lack of expression of a gene in a specific cell due to the non-trivial distinction between true- and false-zero counts. Meanwhile, a false zero is a technical deviation. The RNA-seq technique only detects mRNA molecules that are present, therefore, a gene in the scRNA-seq dataset with a non-zero expression value indicates that at least one mRNA molecule is present. These sparse non-zero expression values provided inspiration for the possibility of performing functional GSEA analysis at the single-cell level.

Functional enrichment analysis may be performed at the single-cell transcriptome expression level given the characteristics of scRNA-seq data. The algorithm is as follows: first, replace the initial expression matrix with a FEM that allows for the direct exploration of the differences between cells from different biological perspectives. Second, the top variable function of a cell cluster obtained by a gene expression matrix (GEM) can be represented as a cell scatter plot. Third, the coverage of a functional gene set is more important than the expression of individual genes for GSEA since genes often overlap different functional groups, which makes single high-expression genes difficult to explain. For example, the activation of a signalling pathway is the result of interactions between all genes; the high expression of one gene does not mean that the pathway is activated.

The current factors affecting the discovery of cell groups are as follows: first, technical covariates must be regressed out before downstream analysis can occur since these factors introduce systematic error and confound the technical and biological variability. This error may lead to systematic differences in gene expression profiles between batches (Leek et al., 2010; Hicks et al., 2018; Chen, Ning & Shi, 2019). The most prominent technical covariates in single-cell data are count depth and batch. The FEM method is based on a gene set, which makes it more robust than the GEM. The measured expression levels for the genes related to a certain biological process may fluctuate between batches or single cells due to variations in sequencing processing. That is, detection for one specific high-expression gene in single cell has a certain randomness. If we can consider the expression of a set of genes related to a specific process, we can increase the detection stability for the biological process of single cells, since the number of expressed genes of a specific biological process obey a hypergeometric distribution. Second, some biological effects may affect the results of the cluster algorithms. For example, the cell cycle phase of a given cell within a cell type population may cause separation from the same cells in the clustering analysis. However, correcting for biological covariates is not always helpful in interpreting scRNA-seq data (Kolodziejczyk et al., 2015). These influencing factors often do not have uniform filtering criteria. In some cases, the cell cycle may be part of the study, or there may be a relationship between the cell cycle and other functions (Haghverdi, Buettner & Theis, 2015; Vento-Tormo et al., 2018; McDavid, Finak & Gottardo, 2016; Blasi et al., 2017). The FEM method can be used to systematically survey the biological aspect of each cell before downstream analysis. To this end, we developed an scRNA-seq FEM algorithm.

Workflow of the proposed methods

The FEM was divided into four steps (Fig. 1). Step 1: The standardized scRNA-seq GEM was transformed into a FEM by multi-module gene enrichment analysis (gene ontology, pathway). Step 2: The p-value represents the significance of enrichment, therefore, the p-value obtained by enrichment analysis was converted into information content (FEM). Step 3: FEM/GEM was used for data standardization, dimensionality reduction clustering, and UMAP visualization (McInnes et al., 2018). Step 4: Differentially expressed genes (DEG) and differentially expressed functions (DEF) analysis was performed based on GEM clustering (see the “GEM and FEM fusion analysis” section for details).

Figure 1 Workflow of the proposed method.

(A) Flowchart of the FEM algorithm, which was used to calculate Fisher’s exact test for each cell and each gene set, following which the calculated p-value matrix was converted into the FEM through information content. The efficiency of the calculation method was improved by matrix operation and multi-core parallel processing (see the Method section for details). (B) Cluster differential expression analysis based on FEM and integration of FEM and GEM (including C and D, the analysis is performed in Seurat). For downstream analysis, FEM can be regarded as another type of GEM. Therefore, Seurat’s analysis pipeline can be used to analyse FEM (C) or treat FEM and GEM as two different omics data for integrated analysis (D). (C) GEM/FEM was used for dimension reduction, clustering, and differential expression genes/function analysis (GC-DEG/FC-DEF). (D) Due to the loss of gene expression information in FEM, the dimensionality reduction and clustering is based on GEM, and differential expression function analysis among clusters in the GEM cluster (GC-DEF) was performed.

Dataset

Peripheral blood mononuclear cells (PBMCs) are populations of immune cells that remain in the less dense upper interface of the Ficoll layer. PBMCs include lymphocytes (T cells, B cells, and NK cells), monocytes, and dendritic cells (DCs). The frequencies of these populations vary across individuals in humans. Lymphocytes are typically in the range of 70–90%, monocytes range from 10–30%, while DCs are only present at 1–2% (Norman, 1995). The PBMCs dataset used in this study was downloaded from the 10X Genomics official website (https://www.10xgenomics.com/resources/datasets), which included B cells, NK cells, CD8 T cells, memory CD4 T cells, naïve CD4 T cells, DC, CD14+ monocytes, FCGR3A+ monocytes, and a small number of platelets. This dataset contained 2,700 cells in total.

The human pancreas dataset contained 2,126 cells and 10 cell types. This included alpha, ductal, endothelial, delta, acinar, beta, gamma, mesenchymal, and epsilon cells, as well as a small number of unknown cell types (Muraro et al., 2016).

The human liver dataset consisted of 777 cells, including seven types of cells: definitive endoderm cells, immediate hepatoblast cells, induced pluripotent stem cells (IPSCs), material hepatocytic cells, hepatic endoderm cells, endothelial cells, and mesenchymal stem cells (Camp et al., 2017).

FEM algorithm

Selected functional groups and their profiles

Three functional gene sets were selected from the Msigdb (v6.2) database for enrichment analysis: the Reactome pathways, gene ontology (GO), and immunologic signature gene sets (Liberzon et al., 2011) (Table 1).

Table 1 Functional gene set.

Function name	Number of gene sets	Details	
C2: Reactome gene sets	1,499	Gene sets derived from the Reactome pathway database.	
C5: GO gene sets	9,996	Gene sets that contain genes annotated by the same GO term. The C5 collection is divided into three sub-collections based on GO ontologies: BP, CC, and MF.	
C7: immunologic signature gene sets	4,872	Gene sets that represent cell states and perturbations within the immune system. The signatures were generated by manual curation of published studies in human and mouse immunology.	

Validation gene set

ImmuneSigDB (Godec et al., 2016) contained a gene set composed of differentially expressed genes from human and mouse immune-related cells. These were collected from the literature with corresponding expression data in the Gene Expression Omnibus (GEO) database. All data were manually reviewed and standardized using the same method. The differentially expressed FDR value was less than 0.02, and the genes were sorted using the mutual information algorithm. No more than 300 genes were selected for each gene set.

Gene-functional group conversion

The cells’ non-zero–expressed genes (i.e., genes in which the read count or normalized expression values, such as RPKM and FPKM, were not zero) in the GEM were extracted first. In the second step, an enrichment analysis score for each cell was calculated. The enrichment analysis method was based on Fisher’s exact test using the Python SciPy package. The Fisher exact test is a statistical test based on a hypergeometric distribution that is used to determine if there are non-random associations between two categorical variables, or to test whether the theoretical value is consistent with the actual value. (1) P=KkN−Kn−kNn

Here, N represents the total number of background genes, which is defined as all non-redundant genes in all gene sets taken from a species’ function database, such as Reactome pathway or GO database. K represents the number of genes in a particular gene set, n represents the number of non-zero genes in a single cell, and k represents the number of genes present in both K and n. The Bonferroni correction was used to counteract the problem of multiple comparisons, but this is optional.

Expression value conversion based on information content

Information content measures the average rate of information from data. The smaller the p-value, the greater the amount of information. For the adjusted p-value of enrichment of a gene set, the null hypothesis is that there is no significant enrichment. So, the smaller the adjusted p-value, the more significant the enrichment of the gene set (rather than stochastic). Therefore, here the information content was used as a measure of the level of expression of a functional group. (2) GSi,j=−logadj−pi,j

Here, i is the ith gene set, j is the jth cell, and adj − pi,j represents the adjusted p-value in the ith gene set in the jth cell.

Algorithm optimization

Fisher’s exact test is a time-consuming process. For single-cell data, a statistical test would be required for each function of each cell. Therefore, when the number of cells is large, the computation time would be untenable. Therefore, the algorithm was optimized with the addition of multi-core computing.

Optimization of the algorithm can be illustrated using the symbols in section “Gene-functional group conversion”. First, because N and K were invariable for all cells, these values were stored in memory to avoid recalculation with each replicate. Second, the gene expression matrix was transformed into 0,1 matrix A, where 1 represents the expression of gene i in cell j and 0 represents the non-expression of gene i in cell j. The gene set was also transformed into 0,1 matrix B, where 1 represents the presence of gene i in set s and 0 represents the absence of gene i in set s. The element (k) in the product A × BT of the two matrices represents the number of genes expressed by cell j in set s (Fig. 2).

Figure 2 Algorithm optimization based on matrix multiplication and multi-core operation.

(A) Convert the GEM and the gene sets into a 0, 1 matrix. The result of multiplication of the two matrices representing the number of genes expressed in cell j and set s. (B) The multi-core parallel computing method established two queues. The data queue was used to store the data needed for calculation and the result queue stored the calculated results. Each set of the two queues uniquely identified the cell and the function to which it belonged. The calculation process adopted a multi-process operation.

Cluster and differentially expressed gene detection based on FEM and integration of GEM and FEM by Seurat

Analysis tools for scRNA-seq data

A number of integrated data analysis software packages and platforms exist, including Seurat (Butler et al., 2018), Scater (McCarthy et al., 2017), and Scanpy (Wolf, Angerer & Theis, 2018). Seurat provides integrated environments (including sample and feature selection, data standardization, dimensionality reduction, clustering, and visualization) to explore massive scRNA-seq datasets (Luecken & Theis, 2019). Seurat was used in this study for normalization, dimensionality reduction, clustering, and visualization.

Dimensionality reduction

The dimensions of single-cell expression matrices were further reduced after feature selection by dedicated dimensionality-reduction algorithms. These algorithms, such as principal component analysis (PCA), embed the expression matrix into a low-dimensional space, which is designed to capture the underlying structure in the data in as few dimensions as possible (Luecken & Theis, 2019; Eraslan et al., 2019). Our data were converted into a linear combination of the first N principal components by the PCA algorithm. The value corresponding to the ‘elbow’ point was taken as the value of N.

Clustering

Single cells were clustered during the analysis of scRNA-seq transcriptome profiles. This may reveal cell subtypes and infer cell lineages based on the relationships among cells. Several software packages support the cluster analysis of scRNA-seq data (Petegrosso, Li & Kuang, 2019). Seurat was used for clustering based on a graphical approach. The cluster function was set to 0.4–1.2 according to the data. All parameters are show in Table S5

Differential expression between clusters

Differential expression analysis is helpful for finding the significant DEG/DEF between distinct subpopulations or groups of cells (Petegrosso, Li & Kuang, 2019). Seurat was used to determine the subsets of functions that exhibited a high variation between clusters.

GEM and FEM fusion analysis

FEM is limited in that it only considers the presence or absence of gene expression without considering the expression value of the gene. Therefore, FEM cannot replace GEM-based methods in cell classification and type identification. In the present study, data from the GEM and FEM were used for fusion analysis (Fig. 1). The multi-modal data analysis module of Seurat was used for polymerization analysis to combine the GEM and FEM results (Stuart et al., 2019; Stuart & Satija, 2019). GEM was used for clustering because it contains information about gene expression values and can better distinguish the differences between cells. FEM reduces all of the genes of the same function to one dimension. If two cells express different genes of the same function, they will be clustered together in FEM, but they may be separated in GEM since they are different genes. We also tested the Differential Expression Functions between Clusters based on GEM (GC-DEF) and the Differentially Expressed Functions between Clusters based on FEM (FC-DEF).

The feature number selection, scaling ratio, PCA, and clustering parameter selection were appropriately adjusted according to circumstances (Table S5), following the Seurat instructions. In brief, we use the VlnPlot visualization function in the Seurat package to select the feature number and scaling ratio value for subsequent analysis. The feature number and the scaling ratio are respectively taken as the maximum value of ordinate that can include all the points after excluding individual excessively high outliers. We can select the number of principal components by observing the corresponding abscissa value at the “elbow” point of the Elbow plot by Seurat. If the number of cell types is known, the clustering parameters were set so that the number of clusters was greater than or equal to the number of actual cell types. If the number of cell types is unknown, we can adjust the number of clusters by setting the resolution parameter of FindClusters function in Seurat. Like most clustering algorithms, there is no uniform standard to define a specific number of clusters. Users can try multiple parameters to get a more appropriate number of clusters. All parameters in this paper are detailed in Table S5.

Evaluation of FEM clustering results

We defined an evaluation score, SC,  to compare the overlap between clusters based on FEM clustering and real cell types. The number of clusters was related to specific parameters, therefore, each cluster should contain only one cell type as much as possible. Additionally, the cluster’s number should be greater than or equal to the number of cell types. Therefore, the clustering parameters were set so that the number of clusters was greater than or equal to the number of actual cell types. For each cluster i, SCi was measured by the consistency of the cell type within the cluster (the proportion of the cell type with the largest number of cells to the total number of cells in the cluster). We used the following formula to evaluate clustering results: (3) SCi=ci,maxci,T

ci,max represents the cell type with the largest number of cells in the i-th cluster. ci,T represents the total number of all cells in the i-th cluster.

Data availability

Publicly available scRNA-seq datasets were used in this study. The PBMCs dataset was downloaded from the 10X Genomics dataset page (https://cf.10xgenomics.com/samples/cell/pbmc3k/pbmc3k_filtered_gene_bc_matrices.tar.gz). The liver and pancreas scRNA-seq data can be accessed in the NCBI Gene Expression Omnibus (GEO) under accession numbers: GSE81252, GSE85241.

Code availability

The FEM software and related scripts have been deposited in the GitHub repository (https://github.com/qingyunpkdd/single_cell_fem).

Results

FEM can separate different cell types

The Reactome pathway gene sets have only 5,741 unique genes and a large amount of gene information was lost when using the FEM algorithm. In comparison GO gene sets had 15,578 unique genes. Therefore, GO gene set-based FEM was used for cluster analysis. The Reactome pathway gene sets were used in the analysis of GC-DEF and their cluster results are shown in Fig. S1. GO-based FEM results are shown in Figs. 3–5.

Figure 3 The comparison between GEM clustering (3A) and GO-based FEM clustering (3B) helps to find the similarities between immune cells.

(A) The results of GEM clustering. (B) The GO-based clustering results. (C) The correspondence between each cluster and cell type of GO-based clustering. (D) The differentiation tree of immune cells (Lim et al., 2013). All clustering results are plotted by UMAP. (C) shows that there are three clusters with low SC scores. The G1 cluster is mainly composed of two types of monocytes (CD14 ± Mono, FCGR3A ±  Mono). It can be seen from (D) that all monocytes have the same recent developmental origin. G2 is composed of NK cells and CD8 T cells. (D) shows that both NK cells and T cells are developed from T/NK cell progenitor cells. G3 is composed of naive CD4 T, CD8 T, and memory CD4 T. The differentiation tree in (D) shows that all T cells have a common nearest ancestor. The No. 7 cluster has a low score; however, it has a small number of cells and the number of cells in all cell types is not the main cluster of the corresponding cell type, and is ignored.

Three data sets (PBMCs, liver, pancreas) were used to replace GEM for dimensionality reduction and clustering to verify whether the GO-based FEM algorithm could separate different cell types. The number of clusters was artificially adjusted and the number of clusters was set to be greater than or equal to the number of actual cell types. This was done so that a cluster contained only one main cell type, which allowed us to better distinguish the different cell types.

We needed to discover different cell groups and determine the relationship between different cell types for single-cell scRNA-seq data. These cells may have similar functions or a common recent differentiation origin. The FEM method may help determine functionally similar cells. Cells were noted to be different subtypes of the same cell if the two groups of cells were far apart in GEM, but close or partially overlapped in FEM. This result may also indicate that the two groups of cells may perform similar functions.

We performed GEM clustering and GO-based FEM clustering on the scRNA-seq data, and compared the two clusters. The FEM clustering results were also compared with real cell types to determine the similarity of cell functions. A low SC score in a particular cluster i indicated that the cluster may contain multiple cell types. A moderate number of a certain cell type A in the cluster i was defined as the main cell type MCi. For a cluster, all MCs may have the same or similar functions.

Figure 4 Comparison of GEM clustering (4A) and GO-based FEM clustering (4B) helps to find the similarities between liver cells.

(A) The results of GEM clustering. (B) The GO-based clustering results. (C) The correspondence between each cluster and cell type of GO-based clustering. (D) The developmental trajectory of stem cells (Camp et al., 2017). All clustering results are plotted by UMAP. (C) shows that a cell group has a low SC score. G1 is mainly composed of definitive endoderm and hepatic endoderm cells. These two types of cells are far apart in GEM (A). But it is close to each other in GO-based FEM cluster. From (D), we find that these two cells are adjacent to each other on the cell differentiation trajectory, and they are the progenitor cells of other cells in the liver.

The results show that the FEM method can distinguish different types of cells that have different functions on three data sets (Figs. 3–5). Figures 3–5 show that FEM can find cells with similar origins or different subgroups of a cell. Figure 3 shows three clusters (G1, G2, G3) that have a low SC score and all of the T cells overlapped in cluster G3. The MCi of NK cells and T cells in cluster G2 literature show that they have a common differentiation origin T/NK cell progenitor. G1’s MCi contained only monocytes (Lim et al., 2013). The FEM in Fig. 4 helped determine the relationship of definitive endoderm cells and hepatic endoderm cells (Camp et al., 2017). Figure 5 illustrates that alpha, beta, gamma, epsilon, and delta cells form one cluster in the FEM clusters, and that the cell differentiation tree verifies that they have a common endocrine progenitor (Jacobson & Tzanakakis, 2017).

Figure 5 Comparison of GEM clustering (5A) and GO-based FEM clustering (5B) helps to find the similarities between pancreatic cells.

(A) The results of GEM clustering. (B) The GO-based clustering results. (C) The correspondence between each cluster and cell type of GO-based clustering and the SC score. (D) The differentiation lineage of pancreatic cells (Jacobson & Tzanakakis, 2017). All clustering results are plotted by UMAP. (C) shows that the two cell groups have the lowest scores. Although G1 has a low score; it has a small number of cells. The number of cells in all cell types is not the main cluster of the corresponding cell type, so it will be ignored. G2 is composed of five types of cells (alpha, beta, delta, gamma, epsilon). Although the epsilon cell has only two cells in cluster 1, but it has only three cells, so it is considered that it is distributed in G2. These five types of cells are far apart in the GEM UMAP plot, but close to each other in the GO-based FEM plot. (D) Shows that all these cells have a common progenitor cell differentiation.

FC-DEF can directly detect the functional differences between clusters

GC-DEF can reveal the expression of specific functions from the UMAP(t-SNE) diagram of GEM and the results of GC-DEF may aid GEM in determining the functions of a specific group. Official data pre-processing in the PBMC dataset included the removal of cells with excessive mitochondrial genes (>5%, quality control) and those with too many (>2,500) or too few (<200) features (Bittersohl & Steimer, 2016). Some cells may have been filtered out in the data pre-processing stage whereas our method did not filter out any cells. Indeed, the filtered cells were found to be located above the NC cells. Using the GC-DEF method, this cell group was found to have a highly expressed cell proliferation-related pathway. Thus, these cells were identified as proliferative (Fig. 6).

Figure 6 An example of GC-DEF functional analysis.

(A) Official clustering results. (B) Results of GEM without cell filtering show a small cell group above the NK cell cluster (when compared with (A). (C) By directly displaying the expression value of specific pathways in all cells, this cell group was found to have high expression of the cell proliferation-related pathway. For downstream analysis, cell screening may result in some cells being filtered out, which may hold some interest. In the process of biological research, cells at special stages often account for only a small part of the total number of cells, but they may be important for explaining biological processes. For example, a small part of cells in the cell cycle stage (B). They are clustered in the NK cell group in clustering, but compared with other NK cells, they are a prominent part of the UMAP plot. When doing differential expression between cell groups, the signals of these proliferating cells will be masked by NK cells. The FEM method is based on a single cell, so it can directly display the functions of these special cells even at single cell level.

The most significantly expressed pathway, “hemostasis”, was located in the platelet clusters. The “innate immune system” pathway was significantly expressed in the monocyte, DC cell, NK cell, and platelet cell populations, which was consistent with literature results (NORMAN, 1995). The top five highly expressed functions were consistent with the cell type characteristics. The other FC-DEF and GC-DEF results are provided in Table S1.

Table 2 shows that most of the corresponding cells were closely related to their corresponding top functions, such as the high expression of “Reactome regulation of beta cell development” in beta cells and the high expression of “Reactome gluconeogenesis” in mature stem cells. All other results are shown in Table S1.

Table 2 Top five pathways for mature hepatocyte cells (liver dataset) and beta cells (pancreas dataset).

Function	Cluster	Adjusted p-value	
reactome-regulation-of-gene-expression-in-beta-cells	beta	7.47E−48	
reactome-regulation-of-beta-cell-development	beta	1.02E−25	
reactome-activation-of-nmda-receptors-and-postsynaptic-events	beta	1.83E−13	
reactome-negative-regulation-of-tcf-dependent-signaling-by-dvl-interacting-proteins	beta	2.05E−11	
reactome-synthesis-of-pips-at-the-early-endosome-membrane	beta	1.76E−06	
reactome-gluconeogenesis	mature hepatocyte	1.75E−11	
reactome-signaling-by-bmp	mature hepatocyte	4.50E−09	
reactome-apoptotic-cleavage-of-cell-adhesion-proteins	mature hepatocyte	9.00E−09	
reactome-transport-of-nucleosides-and-free-purine-and-pyrimidine-bases-across-the-plasma-membrane	mature hepatocyte	1.51E−08	
reactome-bbsome-mediated-cargo-targeting-to-cilium	mature hepatocyte	1.76E−08	

GO-based GC-DEF results was consistent with the pathway-based methods and literature results (Norman, 1995). All other results are presented in Table 3 and Tables S1–S4.

Table 3 Results of the top five GO-based FEM methods for each cluster (PBMC data set).

Function	Cluster	Adjusted p-value	
go-mhc-class-ii-receptor-activity	B	1.31E−299	
go-mhc-class-ii-protein-complex	B	1.94E−207	
go-clathrin-coated-endocytic-vesicle-membrane	B	8.60E−183	
go-clathrin-coated-endocytic-vesicle	B	1.40E−174	
go-mhc-protein-complex-assembly	B	2.39E−173	
go-collagen-containing-extracellular-matrix	CD14+_Mono	3.17E−235	
go-rage-receptor-binding	CD14+_Mono	1.42E−226	
go-chemokine-production	CD14+_Mono	1.61E−211	
go-defense-response-to-bacterium	CD14+_Mono	9.86E−194	
go-neutrophil-migration	CD14+_Mono	2.36E−187	
go-cytolysis	CD8_T	2.61E−64	
go-t-cell-receptor-complex	CD8_T	7.62E−53	
go-negative-regulation-by-host-of-viral-transcription	CD8_T	5.71E−39	
go-regulation-of-cell–cell-adhesion-mediated-by-integrin	CD8_T	1.50E−36	
go-t-cell-receptor-binding	CD8_T	4.19E−29	
go-ige-binding	DC	4.78E−241	
go-t-cell-activation-via-t-cell-receptor-contact-with-antigen-bound-to-mhc-molecule-on-antigen-presenting-cell	DC	6.80E−23	
go-mhc-class-ii-receptor-activity	DC	1.48E−14	
go-hydrolase-activity-acting-on-ester-bonds	DC	0.000373	
go-lipid-metabolic-process	DC	0.000529	
go-igg-binding	FCGR3A+_Mono	9.13E−143	
go-negative-regulation-of-leukocyte-proliferation	FCGR3A+_Mono	4.71E−60	
go-regulation-of-mast-cell-activation	FCGR3A+_Mono	7.62E−59	
go-regulation-of-mast-cell-activation-involved-in-immune-response	FCGR3A+_Mono	1.83E−58	
go-dendritic-cell-differentiation	FCGR3A+_Mono	3.00E−58	
go-positive-t-cell-selection	Memory_CD4_T	5.51E−40	
go-positive-thymic-t-cell-selection	Memory_CD4_T	1.80E−37	
go-t-cell-receptor-binding	Memory_CD4_T	1.89E−37	
go-alpha-beta-t-cell-receptor-complex	Memory_CD4_T	7.89E−35	
go-positive-regulation-of-t-cell-receptor-signaling-pathway	Memory_CD4_T	3.62E−33	
go-t-cell-differentiation-in-thymus	Naive_CD4_T	5.58E−123	
go-thymic-t-cell-selection	Naive_CD4_T	4.23E−87	
go-positive-regulation-of-t-cell-receptor-signaling-pathway	Naive_CD4_T	2.06E−58	
go-t-cell-receptor-complex	Naive_CD4_T	2.18E−57	
go-negative-t-cell-selection	Naive_CD4_T	1.51E−39	
go-granzyme-mediated-apoptotic-signaling-pathway	NK	2.10E−190	
go-cytolytic-granule	NK	1.03E−150	
go-positive-regulation-of-natural-killer-cell-chemotaxis	NK	6.27E−128	
go-cytolysis	NK	6.91E−97	
go-ccr5-chemokine-receptor-binding	NK	9.03E−58	
go-platelet-alpha-granule-membrane	Platelet	6.75E−158	
go-platelet-alpha-granule	Platelet	4.94E−06	
go-platelet-degranulation	Platelet	5.90E−06	
go-platelet-alpha-granule-lumen	Platelet	6.23E−06	
go-contractile-fiber	Platelet	8.58E−06	

Difference between GO enrichment analysis results based on DEG and FEM

The general GSE analysis was based on DEGs between groups (cell types or clusters). The FEM-based gene set (function) enrichment can be defined as the DEF between groups. The results (Table S5, p-value threshold ≤ 0.05) show that: (1) the enrichment analysis results obtained by the two methods are quite different. (2) There are more GO items based on the FEM method than the method based on DEGs.

The possible explanation for the first point is that the two methods are based on different data, and the FEM method is an enrichment result of a single cell. The DEG method is based on results from cell clusters. FEM uses all expressed genes in a cell to calculate GO enrichment, while the DEG method only uses those differentially expressed genes, which may explain why there are more GO items based on the FEM method than the DEGs method. The FEM method focuses on the “differentially expressed GO items”, and the DEG method focuses on “the enriched GO items of differentially expressed genes”.

Validation with an immune dataset

The Immunologic Signatures Collection (ImmuneSigDB) (Liberzon et al., 2011) was used as a validation dataset to test whether the proposed method could detect sets of genes that had been identified as up-regulated or down-regulated by traditional methods. The ImmuneSigDB is composed of gene sets that represent cell types, states, and perturbations within the immune system (Godec et al., 2016). Figure 7 shows that within the bulk RNA-seq dataset, the cell types from the up-regulated expression marker gene set were highly expressed using the method proposed in this study. This demonstrated the efficacy of the proposed method for detecting cell-type-specific gene sets.

Figure 7 Validation results of Immunologic Signatures Collection gene sets.

ImmuneSigDB (Godec et al., 2016) is a collection of gene sets of human and mouse immune cells collected from the GEO database. All data are manually corrected. Each gene set represents genes that are highly expressed in a specific immune cell type compared to another or all other cell types. Here, we verify whether the expression of each gene set (in this case, functional expression) is consistent with the corresponding cell type. (A) UMAP plot for each cell type based on GEM. (B–G) The expression values of various cell type-specific gene sets in each cell. The figures show that the functional expression obtained by our method is consistent with the corresponding cell type.

Discussion

We showed that FEM can be used for cell clustering. It also can merge the GEM method for downstream differential expression analysis to find cell type-specific functions.

It was necessary to evaluate the impact of some biological effects on the cell clusters, such as the impact of cell cycles on cell-type clustering results. Within the same cell type, cells in the cell proliferation cycle and non-proliferating cells may be significantly separated in the cluster and UMAP plots, which affects the final cell type identification and differential expression analysis. In some studies, these cells are removed, while they may still be of interest and there is currently no uniform standard for handling these cells. The method proposed in this study directly converts the expression of genes in cells to the expressions of functions and cells can be screened according to their FEM score at any stage of processing.

It should be noted that FEM only considers the presence and absence of gene expression, without considering the influences of gene-expression values. Therefore, the proposed method and gene-expression–based methods are complementary and should not be considered as alternatives to one another.

We have proven that the FEM-based method can also be applied to cell clustering, and verified the accuracy of cell clustering when compared with real cell types. A direct analysis of the differential expression function between the groups and visually elucidating the differences in specific functions helps to intuitively understand the functions of cells. FEM and GEM were used as two different data sources, while multi-omics data fusion analysis was reused in Seurat and other tools to show the functional results of GEM analysis.

The information that each cell can provide may be valuable for single-cell transcriptome data. The same cell type may also be in different physiological states (for example: different stages of the cell cycle, different metabolic states). Cells in a differentiation stage may have a series of cell states. Therefore, it is necessary to develop data analysis tools for individual cells to find those few but possibly important cells, such as those cells that are in a transient state of differentiation.

Conclusions

The final step in the analysis of gene expression data is to interpret the biological significance of the genes based on a hypergeometric test between the listed genes of interest and the set of genes representing specific functions. If each of the first n genes obtained by differential expression analysis represents a function that does not overlap with the other, then the enrichment analysis will fail. If most genes of a function are expressed in a cell and are screened out in the process of gene selection, this function will also be missed. GSEA at the single-cell level effectively avoids these problems based on the characteristics of single-cell data. The results of the present study showed that direct enrichment analysis at the single-cell level is feasible and yields powerful results.

We preliminarily discussed the applicability of the FEM method and the FEM have a wider range of practical applications. The possible applications of the FEM method include: determining the functional differences between different groups; analysing the function of a few outlier cells; analysing the state of a small number of cells on the cell trajectory in the pseudo-trajectory analysis; analysing which cell groups are more similar in function by comparing GEM and FEM clustering; determining marker quality for different cell groups compared with genes (though there is a need for future validation); and more differential functions were found compared with the GEM method. The disadvantages of the FEM method are: the influence of gene expression is not considered; it is impossible to distinguish cell groups with similar functions; and the function of differential expressions and the result of DEG-based enrichment are quite different, and they cannot be substituted for each other.

Supplemental Information

Supplemental Information 1 Results of Reactome-based FEM cluster

(A, C, E) The GEM clustering results of the PBMC, liver, and pancreas data sets. (B, D, F) The Reactome-based FEM clustering results of the PBMC, liver, and pancreas data sets. There are only 5,741 unique genes in the Reactome gene set. The results showed that many cells in the PBMC and pancreas datasets could not be separated (Table S1 for <!–[if !msEquation]–> <!–[if !vml]–> <!–[endif]–> <!–[endif]–> score).

Click here for additional data file.

Supplemental Information 2 The overlap between clusters of different methods and ground truth cell labels in PBMC (Table S1-1), liver (Table S1-2), and pancreas (Table S1-3) data sets

REACTOME: Reactome pathway-based function cluster and it’s top five most significant pathways. REACTOME MM: gene-based cluster and the top five most significant pathways. GO: Reactome pathway-based function cluster and the top five most significant GO. GO MM: gene-based cluster and the top five most significant GO.

Click here for additional data file.

Supplemental Information 3 Significance of differential expression of all genes and functions between different clusters in the PBMC data set

Table S2-1 Differential expression of all genes between gene-based clusters. Table S2-2 Differential expression of all GO between GO-based clusters. Table S2-3 Differential expression of all GO between gene-based clusters. Table S2-4 Differential expression of all pathways between pathway-based clusters. Table S2-5 Differential expression of all pathways between gene-based clusters.

Click here for additional data file.

Supplemental Information 4 Significance of differential expression of all genes and functions between different clusters in the liver data set

Table S3-1 Differential expression of all genes between gene-based clusters. Table S3-2 Differential expression of all GO between GO-based clusters. Table S3-3 Differential expression of all GO between gene-based clusters. Table S3-4 Differential expression of all pathways between pathway-based clusters. Table S3-5 Differential expression of all pathways between gene-based clusters.

Click here for additional data file.

Supplemental Information 5 Significance of differential expression of all genes and functions between different clusters in the pancreas data set

Table S4-1 Differential expression of all genes between gene-based clusters. Table S4-2 Differential expression of all GO between GO-based clusters. Table S4-3 Differential expression of all GO between gene-based clusters. Table S4-4 Differential expression of all pathways between pathway-based clusters. Table S4-5 Differential expression of all pathways between gene-based clusters.

Click here for additional data file.

Supplemental Information 6 Parameters used in R script

Table S5-1 Parameters in PBMC data set. Table S5-2 Parameters in liver data set. Table S5-3 Parameters in pancreas data set.

Click here for additional data file.

We would like to thank the Southeast University State Key Laboratory of Bioelectronics for providing the computing resources used in our research.

Additional Information and Declarations

Competing Interests

Author Contributions

Data Availability

The authors declare there are no competing interests.

Yunqing Liu conceived and designed the experiments, performed the experiments, analyzed the data, prepared figures and/or tables, authored or reviewed drafts of the paper, and approved the final draft.

Na Lu and Changwei Bi performed the experiments, prepared figures and/or tables, and approved the final draft.

Tingyu Han and Guo Zhuojun performed the experiments, analyzed the data, prepared figures and/or tables, and approved the final draft.

Yunchi Zhu and Yixin Li analyzed the data, prepared figures and/or tables, and approved the final draft.

Chunpeng He and Zuhong Lu conceived and designed the experiments, authored or reviewed drafts of the paper, and approved the final draft.

The following information was supplied regarding data availability:

All source code and data in this article are available at GitHub: https://github.com/qingyunpkdd/single_cell_fem.

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
