# Peer review of "FEM: mining biological meaning from cell level in single-cell RNA sequencing data"

_PeerJ, doi:10.7717/peerj.12570_

## Round 0.1 · original submission · Major Revisions

Dear Dr. Liu,

Your manuscript has been reviewed by two reviewers. Please check their comments and submit the revised manuscript. I agree with reviewer 1 that you should provide a detailed workflow on how to use the FEM method and an example to demonstrate the functionality of the pipeline.

Thanks

·

Basic reporting

The manuscript, on the whole, is straightforward and easy to read; there are a few instances of phrasing which the authors could modify to reduce ambiguity, which have been noted in the PDF as comments. The structure of the paper is suitable, and the figures are legible. However, the figure captions require additional information to clarify the results. These instances have been noted in the PDF.
For bulk sequencing analysis, there have been multiple approaches devised to transform a gene expression matrix to a functional expression matrix at the study-level or single sample-level (e.g. GSVA, ssGSEA, singscore) which the authors clearly appreciate given they have referenced these methods. However, these methods have been applied to scRNA-seq, and further there are additional methods designed specifically for scRNA-seq (e.g. Vision, padog2) to address some of the issues authors detail as motivation for developing their FEM method. It’s clear that the FEM method is distinct from these other methods, however it would be helpful for readers to understand what is unique about the FEM method in the context of other scRNA-seq tools (see Zhang et al. 2020 https://doi.org/10.1016/j.csbj.2020.10.007 for a benchmarking study of these tools applied to scRNA-seq).
Finally, one issue that needs to be addressed is whether the authors intend FEM to replace or complement GEM for downstream analysis. From their methods section and the Github repository, the FEM method is a pathway activity score (PAS) calculation method which does not perform cell clustering, classification, or differential analysis (the authors use Seurat for this). This is not abundantly clear in the introduction and methods, so clarification would be useful.

Experimental design

The manuscript is within the research scope for the journal, and the question they seek to address is relevant to the field. The authors are eager to demonstrate the utility of the FEM approach contrasted with a traditional GEM approach to scRNA-seq analysis, and use public datasets from 3 different tissue sources to highlight how FEM can be used in lieu of GEM and complementing FEM approaches. Benchmarking of PAS schemes for scRNA-seq has been done before, and it’s not in the scope of this type of manuscript to benchmark against other approaches. However, the authors could provide more detail about the results of their comparison between FEM and GEM. Specifically, figures 3D and 4C show overlaps of FEM and GEM clustering/classification, but the captions and figures need more detailed descriptions to help the reader understand the results.
The authors use the terms “FC-DEF” and “GC-DEF” but do not define the acronyms, which makes it difficult to understand what the authors are referring to. The authors state, “ 257 If the two groups of cells are far apart on GEM, but are close or partially overlapped on FEM, it indicates that they are different subtypes of the same cell type, or the two groups of cells may perform similar functions.” This is an interesting proposal, however it isn’t clear from Figure 4 or the text how FEM does this; clarification and expansion of this concept would be very helpful.
With respect to how they use Seurat, the authors state, “The feature number selection, scaling ratio, PCA, and clustering parameter selection were appropriately adjusted according to circumstances” which makes it difficult for readers to recapitulate their analysis. A detailed workflow of 1) how to use the FEM python package on GitHub and 2) how to use Seurat downstream of FEM is needed.
In Figure 6, the authors present reanalysis of bulk RNA-seq datasets from GEO, however the methods for how they arrived at these specific gene sets from the GEO datasets is not in their methods section.

Validity of the findings

The scRNA-seq datasets presented are previously published and publicly available. They have defined their method for FEM calculation and scoring scheme in the methods, use appropriate pathway and ontology genesets (Reactome and GO), and downstream of FEM use appropriate methods for scRNA-seq clustering and differential analysis (Seurat). As stated in the previous section, a description of their Seurat method is needed to fully evaluate their results. The authors figures and results are appropriate for their stated question at the outset. However, the authors’ introduction states that FEM can be a replacement for GEM, while in their discussion, the authors state that FEM is complementary to GEM. I think from their results, the latter is a better description, which would require a modification of the introduction. Further, the discussion section is somewhat short; it would be to the authors’ benefit for them to expand on the utility, simplicity, and broad functionality of the FEM approach. The authors have implemented a classic enrichment test (Fischer’s test) to calculate a pathway activity score (PAS). It’s simple analysis in an elegant fashion and the authors should expand on this further.

Additional comments

In their manuscript “FEM: mining biological meaning from cell level in single-cell
RNA sequencing data”, Liu et al present a method to transform gene-level information to pathway-level information for scRNA-seq clustering and analysis using the classic Fisher’s Exact test. This method can help circumvent one of the main pitfalls of scRNA-seq, namely the sparsity of data with respect to genes annotated. By transforming scRNA-seq data to a Functional Expression Matrix (FEM), the effect missing data (i.e. genes not captured or annotated in a scRNA-seq study) can be minimized by investigating single cell transcriptomics at the pathway or ontology/function level. The authors clearly highlight the rationale for this type of approach to complement a traditional scRNA-seq pipeline demonstrate its utility. While this manuscript and approach is promising, additional information with respect to the 1) methods of analysis following FEM (i.e. clustering and differential expression), 2) comparison between FEM and GEM approaches (e.g. how to interpret overlapping cell cluster assignments in Figure 3D, 4C), and 3) how an FEM approach can aid in the characterization of potential cell-type specific heterogeneity. With further details on the specific methods used downstream of FEM, expanded interpretation and clarification of the results (primarily the complementarity of FEM with GEM), and a longer discussion detailing the method in the context of the wider scRNA-seq field and the utility of the FEM approach, this manuscript would be in a good position for publication at PeerJ.

Reviewer 2 ·

Basic reporting

1) At number of places in Materials and Methods and Results sections, there is spacing errors for references. Authors can correct spacing on Line 100, 101, 103, 123, 156, 270, 278
Line 75, delete symbol “( “ before quality control,
2) In Figure 1: Please correct the spelling error of algorithm in the image 1A
3) In Figure 1C Fene is mentioned. Do the authors mean gene or there is a full name for Fene that is required to be included in manuscript?
The legend of Figure 1C describes FEM, however, authors have not included FEM in the image of Figure 1C
4) On line 194, author states ‘Optimization of the algorithm can be illustrated using the symbols in 2.4.2.’
It is not clear what is 2.4.2 referred to?
5) At number of places, structure and idea stated are hard to understand. Specifically, the sections of Results and Discussion are written with poor editing of English language. There are grammatical errors as well as meaning of the sentence are hard to gauge.
Results section lacks explanation on subsection of figures. Authors needs to explain results in a systematic manner for the better understanding of the data.
In Discussion section, line 293: the sentence meaning is hard to undertand…"Sometimes, evaluate the impact of some biological effects on the cell clusters is necessary, such as the impact of cell cycles on cell-type clustering results"
I suggest if authors can take help from colleague and improve the overall English used in the manuscript.
6) The figure legends are very brief and lack necessary details to understand data. There is scope of describing the legends in a precise and detailed manner. Furthermore, the name of plots ‘t-SNE’ or any other plots as used should explicitly reported in figure legend. Lastly, information on statistical analysis parameters should be included in figure legends of Figure 3, 4, 5, and 6

7) Colors used in Figures 3 and 4 are difficult to read for comparing the data in t-SNE plot and the tabular representation of GO-based FEM cluster. I recommend if the color co-ordination of particular cluster (cell type) can be kept similar in both t-SNE plot and Cell type heading that describes various cell types.
For example acinar cell in Figure 3C image is red, but the color of the same cell in cluster 3 or header of the table is blue.

8) Line 265, NC cells are mentioned. It is not clear what category of cells are being referred NK or DC cells from Figure 5? Authors can please correct this error.

Experimental design

1) The description of idea of using FEM and GEM clusters on line 142, point 4 does not align well with interpretation of same message shown in the pictorial representation of Figure 1 D. Authors needs to clarify how differentially expressed genes are identified using FEM.

2) Although there is a comparative analysis of GO-based FEM cluster with ground truth cell data (Figure 3, 4 and 5), it is not clear what is the detail evaluation metrics used for collecting ground truth cell data versus GO-based FEM cluster. Authors should describe these details in the methods section.

3) As mentioned by the author in line 285, “Figure 6 shows….. method proposed in this study.”
Can authors please clarify the source of the bulk RNA-seq data that is presented in Figure 6 ?
Also, authors can be more specific about the extent to which this validation study overlap with the traditional method they are referring to.
I would also recommend revising the legends of Figure 6 as they lack details on comparison of t-SNE plots in the Figure 6A and 6B.

Validity of the findings

1) Authors have reported t-SNE plots that compare clustering based on GEM and FEM for PBMC dataset in Figure 4, however, they have skipped similar comparison for the data represented in Figure 3.
GEM and FEM based t-SNE plots for liver and pancreas data should be included as a part of comparison of additional cell type in addition of PBMC dataset.

2) The first paragraph and last paragraph of discussion contradicts each other i.e authors claim of using FEM analysis instead of GEM for sc-RNA-seq. There is a scope of improvement of Discussion section.. if authors can structure the ideas in a more systematic manner for giving reader explicitly potential uses and limitations of FEM based algorithm.

3) Line 254: Authors claim that the overlap seen for different cells in PBMC can be interpreted as a functional similarity of some group of cells versus others. Can authors support this claim by describing in detail which cell types showed overlap and how they are functionally related with supporting literature.

Additional comments

Authors have considerably explained the idea of FEM algorithm and tested data-sets from three different sources to test the algorithm. This method eliminate the need of an additional filtering step, which can likely introduce a bias. Furthermore, this methods relies on GEM and FEM based matrix that can be utilized to interpret diverse set of scRNA -seq data in a more meaningful manner. The rationale of this study is promising, however, authors fails to describe many important details that are required for interpretation of the data from this study.

---

## Round 0.2 · Minor Revisions

Dear authors,

The original Academic Editor is unavailable and so I have taken over handling your submission.

Thank you for the thorough revision of your manuscript. There are however a few minor revisions that need to be addressed. Please revise your manuscript and address all the Reviewers' concerns in a point-by point response letter.

Thank you

·

Basic reporting

The reporting of this manuscript is decidedly improved from their first submission. The authors have responded to the comments and critiques from reviewers, and I believe this manuscript meets the reporting guidelines and standards for PeerJ.

Experimental design

The authors have expanded on the description of their FEM algorithm method and included the necessary code to use the algorithm in Github. I believe their FEM method can offer a unique insight to the analysis of sc-RNA-seq data and they have described how their approach fits within the context of the field, and the specific questions it can uniquely address.

Validity of the findings

The authors have used multiple publicly-available datasets to test their method and compared it with a standard method in the field, along with the parameters used for the comparison. This is illustrative of the utility of their method.

Additional comments

The authors have sufficiently and painstakingly addressed all the comments from the reviewers. They have clearly defined the scope, need for, and utility of their FEM method for the analysis of single-cell RNA-seq data. I am very pleased with how they have described their approach within the context of available gene set enrichment methods, and I find the insights gleaned from their approach unique and interesting.

A few minor comments:
- In figures 3, 4, and 5, panel A shows the GEM-based clustering results while panel B shows the FEM-based clustering result. I would recommend clarifying this in the caption for 3B, 4B, and 5B as well as possibly putting "GEM" and "FEM" in some fashion on panels A and B, respectively, for figures 3-5.
- It would be useful to include the GitHub link in the introduction and/or the method description.

Reviewer 2 ·

Basic reporting

Authors have worked substantially towards improving readability of the manuscript with clearly stating applicability of their algorithm for analysis of scRNA-seq data. They have made much clear distinction of how this FEM-based is different from traditionally used GEM-based method.

Minor improvement is recommended in the English editing: There are sentences with excessive use of "however" even in the place where its use is not required.

Experimental design

Lastly, author needs to clarify or clearly state in manuscript how other researcher can use this algorithm and what are the exact parameters used in each software they used. Specifically, the section "GEM and FEM fusion analysis" (starting line 263) is more general and lacks required parameters of the method.

Validity of the findings

-Not Applicable- revised

---

## Round 0.3 · accepted · Accept

Thank you for addressing the minor concerns raised by the reviewers.
Congratulations on the acceptance of your work.

Best regards